# Quantitative and qualitative evaluation of a learning model based on workstation activities

**Judit Sánchez, Cristina Andreu-Vázquez, Marta Lesmes, Marta García-Lecea, Iván Rodríguez-Martín, Antonio S. Tutor, Beatriz Gal**  *

Universidad Europea de Madrid, Madrid, Spain

* beatriz.gal@universidadeuropea.es

**Data Availability Statement:** The data underlying the results presented in the study are included in the Supporting Information.

## Abstract

### Background

Moving towards a horizontal and vertical integrated curriculum, Work-Station Learning Activities (WSLA) were designed and implemented as a new learning instrument. Here, we aim to evaluate whether and how this specific learning model affects academic performance. To better understand how it is received by medical students, a mixed methods research study was conducted.

### Methods

In the quantitative strand, two cohorts of first year students were compared: academic year 2015–2016 n = 320 with no exposure to WSLA, and academic year 2016–2017 n = 336 with WSLA. Learning objectives at different levels of Bloom's taxonomy were identified and performance evaluated from multiple-choice questions. In the qualitative strand, a total of six students were purposely selected considering academic performance and motivation, and submitted to semistructured interviews.

### Results

Performance at both cohorts for learning objectives at lower levels of Bloom's taxonomy was similar (38.8 vs. 39.0%; p = 0.955). In contrast, students in the WSLA group outperformed significantly those not exposed for learning objectives involving upper levels (68.5 vs. 54.2%; p <0.001). A multivariate analysis confirmed that the probability of mastering the second (more complex) objective is 1.64 times higher in students with WSLA methodology (OR 95% CI, 1.15–2.34; p = 0.007) than with traditional methodology. In the interviews, students perceived the clinical scenario of WSLA as a motivator and recognized this methodology as a more constructive framework for understanding of complicated concepts.

### Conclusions

In summary, our mixed methods research supports WSLA as a strategy that promotes deep learning and has a positive impact on academic performance for learning objectives involving higher order thinking skills in medical curricula.

**Funding:** Please note we received funding from the David A Wilson Award. We therefore declare: "The funders had no role in study design, data collection and analysis, decision to publish, or preparation of the manuscript.

**Competing interests:** The authors have declared that no competing interests exist.

## Introduction

Since Flexner´s initial ideas advocating for integration of basic sciences, adopting multidisciplinary approaches in medical education has penetrated most institutions in varying degrees [1]. Thus, integrating basic and clinical sciences has become mainstream in reputed programs [2,3]. Finnerty [4], revisited the standard view to adopt a "basics to clinics" integrated approach by giving special emphasis to the introduction of scientific rigor and basic sciences (anatomy, physiology, biochemistry, etc.) into clinical subjects [5–7]. Here, the concept of horizontal integration was operationally defined as "the organization of teaching material that interrelates or unifies subjects that are taught in separate courses or departments" [8]. In a more vertically integrated curriculum, students are introduced to clinical medicine alongside with basic sciences in the early years of the program [9]. This requires some level of coordination between the different agents involved. [6] Unfortunately, in many universities basic and clinical sciences are still taught unconnectedly across different years and following classical teacher-centered methods. When different basic science departments deliver content in an isolated way, students lack the general view that is required in medical practice [10–12]. They consistently fail to acquire the ability to investigate and analyze diagnosis more holistically. Such an integrated learning model has proven to be more effective for students in treating patients with a unified vision [6].

As suggested by Bloom, successful educational strategies must build across different cognitive levels [13]. While this taxonomy remains popular, other authors such as Webb suggested redefining categories to incorporate levels of complexity in thinking [14]. This brought the focus to the need for aligning learning objectives, activities and assessment [15], and called for learning frameworks that operate over the student engagement and motivation as described by Chi and Wylie [16]. Thus, more effective methodologies were discussed and evaluated to acquire knowledge, skills and competences, as compared with the traditional rather passive learning [17–19]. Students who learn through active learning adopt a constructive approach. They search for an understanding of the subject rather than just reproducing what they heard during lectures. They are encouraged to build knowledge themselves rather than just recalling facts [20]. Active-learning strategies represent an alternative to the uninterrupted lecture to promote a scientific inquiry [21]. These strategies provide opportunities for students to connect abstract ideas to their real-world applications and acquire useful skills, and in doing so to gain knowledge. Such a deeper understanding persists beyond the course experience [22].

To accomplish these goals, we introduced recently Work-Station Learning Activities (WLSA) as a multidisciplinary instrument based on team-based learning model in the Universidad Europea de Madrid (UEM) [23]. Up to that moment, teaching was mostly teacher-centered and based on classical methods. Instead, WSLA serves as an instrument to teach some of the specific learning objectives of basic sciences integrated into a clinical context and therefore promote active and constructive student engagement. Within the WSLA model, the student prepares autonomously the content assigned by the teacher before face-to-face sessions. At the beginning of each session, the student readiness is evaluated through a multiple-choice test (iRAT). During the session, the methodology is organized in workstations, in which each group of 5–6 students function as a team and actively works through the content on their own. Through the completion of each workstation students acquire an integrated understanding of the learning objectives; the instructor´s role is to catalyze the learning process. At the end of the session, a multiple-choice test designed to verify whether students have acquired the initially established learning objectives. In the final evaluation exam at the end of the academic year several questions associated with the learning objectives worked during each WSLA session are included. The WSLA learning model has proven to be effective as a tool for moving towards integrated curricula and students' perception of WSLA is positive in overall terms [23].

In this work, we aimed to evaluate whether the WSLA performed differently as compared with the more traditional approaches running before at the UEM. We aimed to understand whether and how WSLA impacted on academic performance focusing as well on cognitive and motivational aspects of learning. Thus, we conducted a mixed methods research study to analyze whether students' academic results with WSLA outperform traditional learning methodology at different levels, from both cognitive and motivational perspectives. Quantitatively, academic performance was measured in terms of learning objectives acquisition, while qualitatively we analyzed the student´s experience and perception through semi-structured interviews. Our data confirms the strength of WSLA as a new pedagogical strategy within a constructive framework, which facilitates deep learning.

## Materials and methods

Ethics approval and consent to participate was obtained for this study, which was approved by the Universidad Europea de Madrid Research Committee (CIPI/088/17). All participating students signed a document explaining the aims of the study and guaranteeing anonymity, and explicitly declared their consent to publication of results.

### Groups and experimental design

The goal of this study was to analyze whether different cohorts of UEM medical students exhibited different academic performance with our active WSLA methodology (cohort 2016–2017) as compared to traditional approaches (cohort 2015–2016). Inclusion criteria for the medical program is a grade point average above 8 in a scale from 0 to 10 within a Health Science baccalaureate program. Students go through admission procedure with includes a competence test and English language assessment. Both cohorts represent all students from each academic year who met the admission criteria. We aimed to analyze students' experience and their perception of the learning process within the WSLA framework using a qualitative approach. Thus, we adopted a concurrent mixed methods study [24].

The Medical degree program in UEM is the most suitable for the purposes of our research as the study groups have comparable large sizes. The cohorts selected were those of academic year 2015–2016 (n = 320) who received their instruction through traditional methodology versus the academic year 2016–2017 (n = 336) who experienced WSLA methodology (n = 656) first-year medical students).

For the quantitative analysis, we first identified the most suitable learning objectives (LO) that could be compared between WSLA and classical teaching approaches and chose those related to the Blood module. To evaluate student's performance, an evaluating committee identified a set of common multiple-choice question (MCQ) targeting the following LOs:

- LO1: To describe the plasma composition and the different cell types of the blood and its physiological importance.

- LO2: To determine the blood group and demonstrate understanding of the importance of compatibility in blood transfusion.

In the academic year 2015–2016, those LOs were worked through a practical activity following a lecture-based methodology. In the academic year 2016–2017, the same LOs were taught using the WSLA model.

Also, keeping Bloom´s categories in mind [13], we selected these LOs to evaluate effects of different thinking skills; LO1 being a low order thinking skill (LOTS; memorize and remember) and LO2 representing a higher order thinking skill (HOTS; integrate).

We analyzed the answers to MCQs from final semester evaluation. Students' performance was quantified taking into account the percentage of correct, failed and blank questions. To avoid bias in LOs between groups, we homogenized MCQ according to their discrimination, difficulty index, and quality of the questions, for each LO. The difficulty index is a percentage that indicates how difficult the question is as defined by the proportion of incorrectly or blank answers divided by the total number of questions. The discrimination index differentiates the answer of students who obtained the best grades from the answer of students who obtained the worst grades. Using discrimination index, the question is evaluated with an excellent, good, pass or poor quality. Based on these inclusion criteria, we could only choose one question per objective.

For the qualitative strand of this study, we conducted 6 semi-structured interviews with students from the academic year 2017–2018 cohort who experienced the WSLA methodology. They were recruited according to their academic performance (P) (high/low) and motivation (M) (high/low) in their 4 possible combinations (HPHM/HPLM/LPHM/LPLM as identified by consensus among their faculty. Students who accepted to participate in the study fulfilled the following inclusion criteria: HPHM n = 2; LPHM n = 3; LPLM n = 1. Prior to the interview, students were given a brief summary of the research project and their verbal consent to participate was recorded.

## Data collection

For each student within the quantitative strand of the study, the following variables were considered: gender, nationality, language, methodology (classic or WSLA), exam call (first, second. . .), learning objective (LO1 and LO2) and qualifications (from the exam and from the entire subject). The consensus used in order to consider an LO being acquired was a correct answer to the MCQ related to the LO. The consensus used to consider an LO not being acquired was a wrong or a blank answer to the MCQ related to the LO. Personal data was coded to guarantee anonymity. Data analysis has been made with only the variables of students who took the exam. All interviews were audio recorded and transcribed. Personal data was codified during the transcription process to maintain participants' confidentiality. Interviews were in person with researchers guiding the discussion about some concepts considered important for a critical assessment of WSLA.

## Statistical analysis

For the descriptive analysis, the absolute (n) and relative (%) frequencies were used to express the qualitative variables. To test the statistically significant differences of the student's academic results (achievement of LO1 and LO2) in the different cohorts either the Chi-square test or Fisher's exact test was performed for qualitative variables. Multivariate analyses (binary logistic regression models) were built to determine factors affecting accomplishment of each LO. To this purpose, LOs were considered to be achieved or not as an indicator variable, to be associated with methodology (WSLA vs control group), gender and nationality (Spanish versus others). Significance was established at p-value $<0.05$. All analyses were performed using a statistical program STATA IC v.14 (Stata Corp., TX, US).

To analyze students' experience and self-perception of learning with WSLA methodology, Thematic Analysis as described by Clarke and Brown, using Atlas Ti software (version 8) [25] was used.

## Results

Descriptive analysis of both study cohorts is shown in Table 1. From a total of 656 students 596 (90%) sat for the exam and their results were used to assess the academic performance.

**Table 1. Descriptive analysis related to characteristics of 656 first-year medical students, from which were studied Gender and Nationality from Universidad Europea of Madrid of both study cohort 2015–2016 and 2016–2017.**

| Characteristic | Cohort 2015-16(n = 320) | Cohort 2016-17(n = 336) |
|---|---|---|
| **Gender, no. (%)** | | |
| Male | 110 (34.4) | 94 (28.1) |
| Female | 210 (65.6) | 241 (71.9) |
| **Spanish Nationality no. (%)** | 289 (90.3) | 299 (89.3) |
| **Sat for the Exam, no. (%)** | 263 (82.2) | 332 (98.8) |

Interestingly, we observed that proportion of students who took the exam was significantly higher in the WSLA cohort (98.8 vs. 82.2%; $p < 0.001$). Overall, 31.2% of students were men and 68.9% women; 89.8% of Spanish nationality and 10.2% from other nationalities.

From 596 students who took the exam, 231 (38%) achieved LO1. A bivariate analysis was carried out to verify the differences in the proportion of students who reach LO1 for each of the factors, as shown in Table 2. Proportion of students reaching LO1 was similar for each cohort (38.8 vs. 39.0%; $p = 0.955$). Similarly, the rest of the factors did not significantly affect the probability of achieving LO1. We detected a non-significant trend in achieving LO1 for Spanish students versus other nationalities (40.0 vs. 28.1%; $p = 0.078$), although this factor was proven to be independent on the teaching methodology (LO1 achievement in Spanish students: 40.7 vs 39.5%; traditional vs WSLA methodology; $p = 0.762$; LO1 achievement in students from other nationalities: 19.1 vs 33,3%; traditional vs WSLA methodology; $p = 0.362$).

From 596 students who took the exam, 369 (62%) achieved the learning objective 2 (LO2). Table 3 shows the factors affecting the achievement of LO2. The only factor affecting this condition is the methodology (68.5 vs. 54.2%; $p > 0.001$). That is, the proportion of students who achieved LO2 is significantly higher with the WSLA methodology than with the traditional methodology. No other association was found between the rest of the factors and the fact of achieving LO2.

In addition, a multivariate analysis (logistic regression) was used to assess factors potentially affecting the achievement of LO2. This analysis showed that students who receive the WSLA methodology have 1.64 times odds of reaching LO2 than the students receiving the traditional methodology (OR 95% CI, 1.15–2.34; $p = 0.007$).

**Table 2. Bivariate analysis to verify the differences in the proportion of students from Universidad Europea of Madrid, who reach LO1 for each of the factors (Methodology, Gender, Nationality), 2015–2016 and 2016–2017.**

| Characteristic | Students who achieve LO1 (n = 320) | | P-value |
|---|---|---|---|
| | Number | (%) | |
| **Methodology** | | | |
| WSLA | 128 | 38.8 | 0.950 |
| Traditional | 103 | 39.0 | |
| **Gender** | | | |
| Male | 74 | 41.3 | 0.421 |
| Female | 157 | 37.8 | |
| **Nationality** | | | |
| Spanish | 215 | 40.0 | 0.078 |
| Other | 16 | 28.1 | |

*P value to find significant differences ($p < 0.05$) between cohorts.

**Table 3. Bivariate analysis to verify the differences in the proportion of students from Universidad Europea of Madrid, who reach LO2 for each of the factors (Methodology, Gender, Nationality), 2015–2016 and 2016–2017.**

| Characteristic | Students who achieve LO2 (n = 336) | | P-value |
| --- | --- | --- | --- |
| | Number. | (%) | |
| **Methodology** | | | |
| WSLA | 226 | 68.5 | < 0,001* |
| Traditional | 143 | 54.2 | |
| **Gender** | | | |
| Male | 113 | 63.1 | 0,740 |
| Female | 256 | 61.7 | |
| **Nationality** | | | |
| Spanish | 330 | 61.5 | 0,302 |
| Other | 39 | 68.4 | |

*P value to find significant differences (p < 0,05) between cohorts.

Regarding the qualitative analysis, using personal interviews allowed us to identify five major themes in which to focus the analysis. Three of them are based on theoretical perspective and were the focus of our investigative approach (deductive themes: teaching methodology, integration and learning process). Two more emerged from the data, thus inductive, and proved to be very relevant to the study (role of the student and the professor).

Concerning the teaching methodology, their opinions were positive especially with respect to their motivation and the social interaction through group work dynamics.

*"I learnt a lot with integrated activities, and one is not so aware of it until you sit for the exam"* (M2 3:25).

*". . .workstations motivate a lot. . . as they are different things, you work with different stuff; maybe with four workstations you have worked four little things but you will retain them"* (M4 10:29).

*"Being organized in groups, favors team work"* (M3 9:18).

However, students would value it as much as other active learning methodologies that they are usually exposed to. Moreover, students appreciated a combination of traditional lectures and active learning methodologies, including WSLA. Lectures were an element that students feel is necessary.

*"I would recommend a traditional lecture before one of these activities, meaning that they ("the activities") are not a substitute but a complement".* (M3 9:13).

*"It is true that the lecture, plus you autonomous reading and the activity is like the perfect triad that helps me, and I enjoy"* (M3 9:5)

When considering the concept of integration and the learning process itself, students recognized the benefit of integration as an identity trait of UEM. They stressed two aspects linked to this theme when specifically referring to WSLA. On one hand, the importance of the clinical context as framework for WSLA, supports the idea of the motivational aspect of vertical integration in Medicine. On the other hand, they pointed to the fact that a combination of different basic sciences facilitates the learning process through a horizontal integration.

Interestingly, when discussing the concept of integration, students wished they could have integrated activities more often.

> *"I believe the added value of the university is the integration of basic sciences, which occurs during the basic sciences years"* `(M1 2:23).`

> *"As an aggregated value, aiming to integrate several basic sciences always with a clinical point of view, is the interesting aspect of the integrated activities"* (M1 _2:28).

> *"I would have loved to have more of these activities"* (M3 9:31).

In relation to the previous theme, horizontal and vertical integration within the WSLA helped students with concept retention and facilitates understanding of complex ideas. This content may have been introduced in a lecture but exposure to WSLA scenarios facilitates understanding. This finding supports the results obtained in the Quantitative strand of the study and reinforces the value of WSLA towards facilitating the understanding of complex content

> *"Knowing something to sit for an exam is one thing, and another is to learn it [. . .] and the integrated activity has helped me"* (M1 2:3).

> *"It is a lot easier to comprehend everything"* (M5 11:1)

Importantly, two additional themes emerged in the interviews. First, the role of the professor and their relationship with students emerged as a key element for the learning process. Students identified knowledge and expertise as cultural aspects, as well as teaching methodology.

> *"Lecturers clarify misconceptions, they open a space for dialogue and it is very nice"* (M3 9:21)

> *"Lecturers give feedback to the class at the end of the activity"* (M3 9:18).

Secondly, students and their own motivation, aspirations and ability to work in a group were also explicitly emphasized. The time given in each workstation of the WSLA emerged recurrently through the interviews and interestingly with opposite reflections. While for some students, the short time given was perceived as a benefit of the methodology because it forces them to be focused and prepared beforehand; others saw it as a drawback of the methodology that did not permit a full comprehension of the concepts.

> *"because it is not only you answering to the activity, it is also that one is exchanging information with other students and listening to their opinions–which is something I value greatly– and our lecturers give us feedback at the end.* (M3 9:18).

> *"I think it is important to be prepared in advance as we do not have all the time in the world [. . .] I believe in order to make the most out of the practical session, the work you do beforehand is very important"* (M2 3:23).

## Discussion

Here, we evaluated quantitatively and qualitatively the WSLA as a learning model. Our findings show that the percentage of WSLA trained students who decided to go to the final exams was significantly higher when compared with those experiencing traditional methods. We also find that WSLA is a suitable tool for the students to meet LOs at different cognitive levels as

defined by Bloom's taxonomy. Although no significant difference emerged at the bottom of the pyramid (remember), we find that WSLA allowed the students to meet LOs that are categorized at a superior level in the cognitive domain of Bloom´s taxonomy (understand and apply). This suggests that WSLA may have a stronger impact at higher integrative levels, thus allowing for more directed pedagogical strategies in medical education.

Medical professionals are expected to integrate content that is traditionally taught in isolation, and this may be particularly challenging in the context of clinical practice. Professional competence involves the capacity to autonomously and effectively use all resources acquired during formative years (*both content and skills*). We feel it is critical for medical students to experience integrated concepts before they first interact with patients. WSLA is a suitable methodological tool to facilitate and encourage this type of learning. Horizontal integration in the format of WSLA helps students with concept retention and facilitates understanding of complex ideas. This content may have been introduced in a lecture but exposure to WSLA scenarios facilitates understanding.

In previous work, we described that one of the advantages of moving into the integration of Basic Sciences with WLSA is the benefit to provide medical education with a holistic and less fragmented approach [23]. A better understanding of foundational courses in a clinically relevant context may improve students' performance and employability, since concepts from basic disciplines are essential today for understanding and treating diseases. The WSLA instrument better trains students to solve clinical cases, which is highly motivating for them especially in early years [23]. Students with higher grade point averages, who studied for longer hours and reported to be more motivated to succeed, perform better academically [26]. From this perspective, positive drives, such as task-related enjoyment used in WSLA, lead to greater interest and intrinsic motivation [27,28]. Students consistently agreed upon the idea that the clinical scenario during WSLA is motivating, supporting the concept of vertical integration in Medicine. This reinforces WSLA as a learning tool that enhances intrinsic motivation to succeed academically. Motivated students tend to believe they are better prepared and this could be at the root of the significant difference we found in the achievement of LOs.

It is also important to consider how emotions associated to learning can influence to increase extrinsic motivation as well; that is, students' motivation to engage in academic tasks as a means to reach their final objective [29]. While there is some controversy on this matter [26] and extrinsic motivation is considered by most psychologists to be less beneficial and long lasting than its intrinsic counterpart [27], it may act to stimulate health science students to achieve good academic performance, well-being and satisfaction in their professional career. This factor is especially important in learning basic sciences such as anatomy, histology and physiology in health science curricula [30]. In fact, the WSLA methodology is considered critical to motivate the students' learning process in foundational sciences, as we have recorded in the analyzed interviews. Students and their own motivation, aspirations and ability to work in a group is explicitly emphasized: *"I believe student´s motivation plays a very active role in Medicine [. . .] the methodology used by the lecturer is essential, but I think more than anything it is important how much you study and how motivated you are" (M1 2:21)*. Notably, similar to other learning frameworks such as ICAP [15, 31], our analysis reveals WSLA as an important instrument in bringing students' to higher levels of engagement by promoting interactions with others: *"because it is not only you answering to the activity, it is also that one is exchanging information with other students and listening to their opinions–which is something I value greatly–and our lecturers give us feedback at the end.* (M3 9:18).

It is important to highlight that the multivariate analyses identified WSLA as the only variable affecting the acquisition of a particular LO for first year medicine students. Lorin Anderson, a former Bloom's student, revised his taxonomy to add relevance and guidance for 21$^{st}$

century teachers and learners [32]. To determine if the first level of Bloom´s Taxonomy (remember) is reached, questions are asked solely to test whether students have gained specific information from lessons and remembered some facts and data of the main ideas being taught. In our study, LO1 belongs to this category because questions were made to test whether students had memorized the plasma composition and different cell types of the blood and whether they remembered the physiological importance of cellular types.

In the next two levels of Bloom's Taxonomy, i.e. understand and apply, students have to process the information and apply it to a real context. Within LO2, instead of simply being able to name the various types of cellular blood types, students are asked to understand blood typing and they are asked to simulate an emergency scenario. Therefore, the instructor has to define the assessment questions for this level of thinking. Application questions are those where students have to actually apply, or use, the knowledge they have learned. They might be asked to solve a problem with the information they have gained in class being necessary to create a viable solution. For example, related to the LO2, a student might be asked to solve what type of blood he would use to transfuse to a patient who runs into the emergency room. A successful teaching session occurs when students have positively met the learning objectives of the lesson and are able to easily comprehend what they have been taught. We have found that the proportion of students meeting LO2 with WSLA is significantly higher than those who studied through the traditional methodology.

At this point it is relevant to reflect on the contributions by previous authors on critical learning strategies. Two approaches have been identified in this respect: surface and deeper or more complex ones [33,34]. Looking at the surface approach, the intention is just to become comfortable with the content and it sees the course as unrelated bits of information leading to a more restricted learning process such as routine memorization [35]. In contrast, in a deeper approach the intention is to extract meaningful results through an active learning process that involves the ability to relate ideas and to look for patterns and principles with a more holistic strategy [33,36]. However, the complexity of the individual ways of learning is infinite, so it is neither ethical nor appropriate to ascribe students into a single learning category. Any given approach can only be applied with confidence to a particular teaching-learning environment, as that given approach will be the result of an interaction between the student and the context [35]. In this sense, we can say that the learning process to reach LO1 needs a surface learning strategy, based on memorization in which the learning methodology is not as relevant as to reach the LO2. Similarly, LOs can be considered under the view of Webb's depth of knowledge and be seen in terms of evolution across a more broadly defined cognitive spectrum [14].

Interestingly, we observed that there is a higher proportion of Spanish students meeting LO1 as compared with other nationalities. This is probably due to their educational background because the Spanish educational system is mainly centered and based on reproducing and acquiring information [37]. Although we found this result, there is no difference in reaching LO1 depending on the classical versus WSLA methodology.

WSLA is a student-centered methodology, designed to promote integrative knowledge, interpretations and calculations using workstations and clinical scenarios. Importantly, the last step of each WSLA module leads to a final simultaneous report of the clinical scenario wrapped up in a final debriefing. These features of the WSLA model are more suitable for applying knowledge and skills, for a deep learning strategy and for a transformative learning process. Indeed, it is been reported that the mnemonic benefits of testing are further enhanced by feedback, which helps students to correct errors and confirm correct answers [38]. The importance of having a teacher in the classroom setting, who can provide adequate mentorship, direct supervision, and immediate feedback to help engage learners, cannot be underestimated. In the WSLA protocol, the teacher supervises all the learning process acquiring a

secondary very important role as our students have reported with their verbatums: *"Lecturers clarify misconceptions, they open a space for dialogue and it is very nice"* (M3 9:21). We feel our work supports exploiting WSLA as a pedagogical strategy to drive medical students up to the superior levels in the cognitive spectrum.

## Conclusions

In summary, our quantitative and qualitative analyses support WSLA as a teaching and learning methodology has a positive impact on academic performance for learning objectives in the higher order thinking skills category. Moreover, WSLA is perceived as a motivating strategy by students which also facilitates deeper learning to better engage medical students in their own learning process.

## Supporting information

**S1 Table. This is an anonymized excel file with all data used in this manuscript.**
(XLSX)

## Acknowledgments

We would like to acknowledge Dra. Ana Isabel Rodríguez-Learte and Dra. Mª del Rocío González for their support and ideas. In addition, we thank to our entire research team for joint work.

## Author Contributions

**Conceptualization:** Marta Lesmes, Antonio S. Tutor, Beatriz Gal.

**Data curation:** Judit Sánchez.

**Formal analysis:** Judit Sánchez, Cristina Andreu-Vázquez, Marta Lesmes.

**Funding acquisition:** Beatriz Gal.

**Investigation:** Judit Sánchez, Cristina Andreu-Vázquez, Marta García-Lecea, Antonio S. Tutor, Beatriz Gal.

**Methodology:** Marta Lesmes, Marta García-Lecea, Iván Rodríguez-Martín.

**Supervision:** Beatriz Gal.

**Validation:** Beatriz Gal.

**Writing – original draft:** Iván Rodríguez-Martín, Beatriz Gal.

**Writing – review & editing:** Judit Sánchez, Marta Lesmes, Marta García-Lecea, Iván Rodríguez-Martín, Antonio S. Tutor, Beatriz Gal.

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
