## [Decision Letter · Decision Letter 0]

25 Jun 2020

PONE-D-20-03757

Quantitative and qualitative evaluation of a learning model based on workstation activities

PLOS ONE

Dear Dr. Gal,

Thank you for submitting your manuscript to PLOS ONE. After careful consideration, we feel that it has merit but does not fully meet PLOS ONE’s publication criteria as it currently stands. Therefore, we invite you to submit a revised version of the manuscript that addresses the points raised during the review process.

Currently the paper does not meet even the lowest level of criteria for publication in PLoS One. Most specifically there is no clear question that is addressed by the research. This means that the paper needs a very extensive rewrite before it will be suitable for publication. To start with there needs to be a clear explanation of the expected differences that you think will occur because of the use of the work-station based methods. The introduction needs to be strengthened with a deeper look at theory and in particular current methods of teaching and extrinsic and intrinsic motivation theory as well as Bloom and extensions from Bloom. If you are not able to meet the publication criteria in a revised manuscript then it may be rejected at that stage. I have attached a copy of the manuscript with some more detailed comments.

We look forward to receiving your revised manuscript.

Kind regards,

Andrew R. Dalby, PhD

Academic Editor

PLOS ONE

Journal Requirements:

2. PLOS ONE will consider submissions that present new methods, software, or databases as the primary focus of the manuscript if they meet the criteria of utility, validation, and availability described here: http://journals.plos.org/plosone/s/submission-guidelines#loc-methods-software-databases-and-tools. To meet these criteria, please provide supporting materials enabling other teachers and researchers to replicate your teaching intervention such as sample worksheets, a detailed lesson plan or curriculum or other educational materials. If you include supporting materials, they should not be under a copyright more restrictive than CC-BY.

"This work was funded by the 2017-2018 Wilson Award and the 2018/UEM20"

"The author(s) received no specific funding for this work"

5. Your ethics statement must appear in the Methods section of your manuscript. If your ethics statement is written in any section besides the Methods, please move it to the Methods section and delete it from any other section. Please also ensure that your ethics statement is included in your manuscript, as the ethics section of your online submission will not be published alongside your manuscript.

Reviewers' comments:

Reviewer's Responses to Questions

**Comments to the Author**

1. Is the manuscript technically sound, and do the data support the conclusions?

Reviewer #1: No

2. Has the statistical analysis been performed appropriately and rigorously? 

Reviewer #1: Yes

3. Have the authors made all data underlying the findings in their manuscript fully available?

Reviewer #1: Yes

4. Is the manuscript presented in an intelligible fashion and written in standard English?

Reviewer #1: Yes

5. Review Comments to the Author

Reviewer #1: The authors describe a learning model that by modern educational research standards would be obviously better than pure lecture and the benefit would show up to a greater extent on more conceptually difficult to answer questions, unless technical details were poorly done. There is not an open research question to answer here. This is a quality assurance project (were these materials well designed at the details level).

While Bloom's Taxonomy remains popular in applied contexts, it is terribly outdated. The full taxonomy has many levels beyond just higher and lower as a binary. There never was a clear empirical basis for the original taxonomy, nor has there been a systematic validation of it or the revised one. "Apply" is a hopelessly broad category. Webb's Domains of Knowledge could be used, since it actually has some empirical support. If one wants to use a general framework from this century, I would suggest Chi and colleagues' ICAP framework. Easy and clear in how to use it in research; strong empirical validation.

As an empirical study, there were also many problems. The two cohorts were not checked for equivalence on incoming standards. Focusing on just one model is thin from a generalizability standpoint. Using one multiple choice question is terrible from a measurement perspective. The interview analysis had a clear bias of interpretation: when the students asserted that the combination of lecture and workstation was important, the authors assumed (ironically!) that the students were wrong and their answers reflected cultural bias.

6. PLOS authors have the option to publish the peer review history of their article (what does this mean?). If published, this will include your full peer review and any attached files.

Reviewer #1: No

---

## [Author Response · Author response to Decision Letter 0]

15 Jul 2020

Reviewer #1: The authors describe a learning model that by modern educational research standards would be obviously better than pure lecture and the benefit would show up to a greater extent on more conceptually difficult to answer questions, unless technical details were poorly done. There is not an open research question to answer here. This is a quality assurance project (were these materials well designed at the details level).

We apologize our research question was not clearly posed in the previous version. Please note we introduced the WSLA in 2016, as a flexible learning instrument in our university. Up to that moment, most teaching followed classical methodology and basic and clinical sciences were poorly integrated. Thus, our main motivation with this work was to evaluate quantitative and qualitatively whether and how WSLA affects academic performance and to understand student´s experience with the methodology. We found WSLA useful to integrate learning at the higher cognitive level while promoting student engagement. We have revised the manuscript to make our question clearer. See Abstract (first paragraph) and Introduction (last paragraph).

While Bloom's Taxonomy remains popular in applied contexts, it is terribly outdated. The full taxonomy has many levels beyond just higher and lower as a binary. There never was a clear empirical basis for the original taxonomy, nor has there been a systematic validation of it or the revised one. "Apply" is a hopelessly broad category. Webb's Domains of Knowledge could be used, since it actually has some empirical support. If one wants to use a general framework from this century, I would suggest Chi and colleagues' ICAP framework. Easy and clear in how to use it in research; strong empirical validation. 

We thank the reviewer for bringing our attention to this very important point. We feel that in the context of our study, Bloom’s and Webb’s definitions apply in relation to learning goals and assessment. The empirical character of the learning framework is another aspect to consider. We appreciate the reviewer suggestions to rely on the ICAP framework, but because our practice was focused on learning objectives, our study adopted Bloom´s taxonomy. Notably, we don’t see Bloom´s taxonomy as binary. The fact that we identified difference at low and high levels rather reflects the effect in the continuum of the cognitive spectrum. 

However, we feel that incorporating the reviewer´s perspectives benefits the analysis of our work. Indeed, the WSLA as a flexible learning instrument allows combining multiple perspectives. Thus, we have substantially edited the Introduction to connect the evolution from Bloom’s and Webb’s considerations into more constructive and empirically demonstrated frameworks, such as ICAP. Later, at the Discussion, we revolve around these ideas to extend the reach of our conclusion further and we suggest indeed that adopting ICAP is desirable for a future work. We feel the manuscript has gained a wider view. See Introduction and Discussion. 

As an empirical study, there were also many problems. The two cohorts were not checked for equivalence on incoming standards. 

We appreciate this comment. We have added information on incoming standards in the revised version. See page 5.

Focusing on just one model is thin from a generalizability standpoint. 

We understand the reviewer meant the WSLA model. Please, note that our motivation was to assess this model versus the traditional. Of course, we are running different programs and models in our everyday health science education. Those are now under review in other journals. We feel it is important for clarity sake to address them one by one. 

Using one multiple choice question is terrible from a measurement perspective. 

Indeed, we had many multiple choice questions to measure each learning objective. However, we included only those questions meeting statistical criteria regarding similar difficulty and discrimination index and quality of questions in the two cohorts to avoid confounding factors. While this was specified in Materials and Methods of the previous version, we have revised the current version to make even clearer. See page 6.

The interview analysis had a clear bias of interpretation: when the students asserted that the combination of lecture and workstation was important, the authors assumed (ironically!) that the students were wrong and their answers reflected cultural bias.

We fully agree. Thank you for highlighting this. This was removed in the masnucript accordingly.

---

## [Editor Report · Decision Letter 1]

17 Jul 2020

Quantitative and qualitative evaluation of a learning model based on workstation activities

PONE-D-20-03757R1

Dear Dr. Gal,

We’re pleased to inform you that your manuscript has been judged scientifically suitable for publication and will be formally accepted for publication once it meets all outstanding technical requirements.

Kind regards,

Andrew R. Dalby, PhD

Academic Editor

PLOS ONE
---

## [Editor Report · Acceptance letter]

23 Jul 2020

PONE-D-20-03757R1 

Quantitative and qualitative evaluation of a learning model based on workstation activities 

Dear Dr. Gal:

I'm pleased to inform you that your manuscript has been deemed suitable for publication in PLOS ONE. Congratulations! Your manuscript is now with our production department. 

Kind regards, 

on behalf of

Dr. Andrew R. Dalby 

Academic Editor

PLOS ONE